# Network Calibration under Domain Shift based on Estimating the Target Domain Accuracy

## Abstract

In this study, we address the problem of calibrating network confidence while adapting a model that was originally trained on a source domain to a target domain using unlabeled samples from the target domain. The absence of labels from the target domain makes it impossible to directly calibrate the adapted network on the target domain. To tackle this challenge, we introduce a calibration procedure that relies on estimating the network's accuracy on the target domain. The network accuracy is first computed on the labeled source data and then is modified to represent the actual accuracy of the model on the target domain. The proposed algorithm calibrates the prediction confidence directly in the target domain by minimizing the disparity between the estimated accuracy and the computed confidence. The experimental results show that our method significantly outperforms existing methods, which rely on importance weighting, across several standard datasets.

## 1 Introduction

Deep Neural Networks (DNN) have shown remarkable accuracy in tasks such as classification and detection when sufficient data and supervision are present. In practical applications, it is crucial for models not just to be accurate, but also to indicate how much confidence users should have in their predictions. DNNs generate confidence scores that can serve as a rough estimate of the likelihood of correct classification, but these scores do not guarantee a match with the actual probabilities (Guo et al., 2017). Neural networks tend to be overconfident in their predictions, despite having higher generalization accuracy, due to the possibility of overfitting on negative log-likelihood loss without affecting classification error (Guo et al., 2017; Lakshminarayanan et al., 2017; Hein et al., 2019). A classifier is said to be calibrated with respect to a dataset sampled from a given distribution if its predicted probability of being correct matches its true probability. Various methods have been introduced to address the issue of over-confidence. Network calibration can be performed in conjunction with training (see e.g. (Mukhoti et al., 2020; Müller et al., 2019; Zhang et al., 2022)). Post-hoc scaling methods for calibration, such as Platt scaling (Platt et al., 1999), isotonic regression (Zadrozny & Elkan, 2002), and temperature scaling (Guo et al., 2017), are commonly employed. These techniques apply calibration as post-processing, using a hold-out validation set to learn a calibration map that adjusts the model's confidence in its predictions to become better calibrated.

The implementation of deep learning systems on real-world problems is hindered by the decrease in performance when a network trained on data from one domain is applied to data from a different domain. This is known as the domain shift problem. There are two main types of domain shift; namely, covariate shift (Sugiyama et al., 2007) and label shift (Lipton et al., 2018). In this study, we focus on covariate shift, a scenario where the distribution of features changes across domains, but the distribution of labels remains constant given the features, since it is more frequent in recognition tasks. In an Unsupervised Domain Adaptation (UDA) setup we assume the availability of data from the target domain but without annotation. There is a plethora of UDA methods that are based on strategies such as adversarial training methods that aim to align the distributions of the source and target domains (Ganin et al., 2016), or self-training algorithms that are based on computing pseudo labels for the target domain data (Zou et al., 2019).

Table 1: Comparison of calibration methods for unsupervised domain adaptation (UDA).

| Calibration Method | Designed to domain shift | Works without target label | Works on target data | Approach | Granularity |
|---|---|---|---|---|---|
| Temp. Scaling (Guo et al., 2017) | × | × | × | – | Instance level |
| CPCS (Park et al., 2020) | ✓ | ✓ | × | Importance weight estimation | Instance level |
| TransCal (Wang et al., 2020) | ✓ | ✓ | × | Importance weight estimation | Instance level |
| UTDC (proposed) | ✓ | ✓ | ✓ | Estimates target accuracy | Dataset level |

In this study we tackle the problem of calibrating predicted probabilities when transferring a trained model from a source domain to a target domain without any given labels. Studies show that present-day UDA methods are prone to learning improved accuracy at the expense of deteriorated prediction confidence (Wang et al., 2020). Calibrating the confidence of the adapted model on data from the target domain is challenging due to the coexistence of the domain gap and the lack of target labels. Current calibration methods that are applied to the adapted model use the labeled validation set from the source domain for calibration (Park et al., 2020; Wang et al., 2020; Pampari & Ermon, 2020). They apply Importance Weighting (IW) to correct the shift from the source to the target by assigning higher weights to source examples that resemble those in the target domain. In practice, even after the domain adaptation process, the accuracy on the source domain where labels are available, is greater than the accuracy on the target domain. Hence, the accuracy estimation when calibrating the target domain using the source data is too optimistic. Calibrating neural networks is necessary because they are often overconfident in their predictions compared to their actual accuracy (Guo et al., 2017; Lakshminarayanan et al., 2017; Hein et al., 2019). If the accuracy is overestimated, it conceals the overconfidence issue, leading to a suboptimal temperature scaling value in the case of temperature scaling. Another drawback of IW methods is that they only use the unlabeled target data to train a binary source/target classifier that is used to set the weights of the source samples. However, the network confidence is independent of the true labels and can thus be directly computed on the target data.

We propose a UDA calibration method that computes the confidence and estimates the accuracy directly on the target domain. We first assess the accuracy in the target domain. Then we find calibration parameters that minimize the Expected Calibration Error (ECE) measure (Naeini et al., 2015) on the target domain. A comparison of typical calibration methods is shown in Table 1. Our major contributions include the following:

- We show that importance weighting relies on an overly optimistic estimation of the target accuracy and thus is not relevant for large covariate shift.

- We propose a calibration method that is directly applied to the target domain data, based on a realistic estimation of the accuracy of the adapted model on the target domain.

We evaluated our UDA calibration algorithm on several standard domain adaptation benchmarks. The results of our approach on all benchmarks consistently outperform previous works, creating a new standard of calibrating networks for unsupervised domain adaptations.

## 2 BACKGROUND

Consider a network that classifies an input image $x$ into $k$ pre-defined categories. The last layer of the network is comprised of $k$ real numbers $z = (z_1, ..., z_k)$ known as *logits*. Each of these numbers is the score for one of the $k$ possible classes. The logits are then converted into a soft decision distribution using a *softmax* layer: $p(y = i|x) = \frac{\exp(z_i)}{\sum_j \exp(z_j)}$ where $x$ is the input image and $y$ is the image class. Despite having the mathematical form of a distribution, the output of the softmax layer does not necessarily represent the true posterior distribution of the classes, and the network often tends to be over-confidenct in its predictions (Guo et al., 2017; Lakshminarayanan et al., 2017; Hein et al., 2019). The predicted class is calculated from the output distribution by $\hat{y} = \arg\max_i p(y = i|x) = \arg\max_i z_i$. The network *confidence* for this sample is defined by $\hat{p} = p(y = \hat{y}|x) = \max_i p(y = i|x)$. The network *accuracy* is defined by the probability that the most probable class $\hat{y}$ is indeed correct. The network is said to be *calibrated* if the estimated confidence coincides with the actual accuracy.

The Expected Calibration Error (ECE) (Naeini et al., 2015) is the standard metric used to measure model calibration. It is defined as the expected absolute difference between the model's accuracy and its confidence. In practice, the ECE is computed on a given validation set $(x_1, y_1), ..., (x_n, y_n)$. Denote the predictions and confidence values of the validation set by $(\hat{y}_1, \hat{p}_1), ..., (\hat{y}_n, \hat{p}_n)$. To compute the ECE measure we first divide the unit interval $[0, 1]$ into $M$ equal size bins $b_1, ..., b_M$ and let $B_m = \{t | \hat{p}_t \in b_m\}$ be the set of samples whose confidence values belong to bin $b_m$. The network average accuracy at this bin is defined as $A_m = \frac{1}{|B_m|} \sum_{t \in B_m} \mathbb{1}(\hat{y}_t = y_t)$, where $\mathbb{1}$ is the indicator function, and $y_t$ and $\hat{y}_t$ are the ground-truth and predicted labels for $x_t$. The average confidence at bin $b_m$ is defined as $C_m = \frac{1}{|B_m|} \sum_{t \in B_m} \hat{p}_t$. If the network is under-confident at bin $b_m$ then $A_m > C_m$ and vice-versa. The ECE is defined as follows:

$$\text{ECE} = \sum_{m=1}^{M} \frac{|B_m|}{n} |A_m - C_m|. \tag{1}$$

The ECE is based on a uniform bin width. If the model is well-trained, most of the samples should lie within the highest confidence bins. Hence, the low confidence bins should be almost empty and therefore have no influence on the computed value of the ECE. For this reason, we can consider another metric, Adaptive ECE (adaECE) where the bin sizes are calculated so as to evenly distribute samples between bins (Nguyen & O'Connor, 2015):

$$\text{adaECE} = \frac{1}{M} \sum_{m=1}^{M} |A_m - C_m| \tag{2}$$

such that each bin contains $1/M$ of the data points with similar confidence values.

Temperature Scaling (TS), is a standard, highly effective technique for calibrating the output distribution of a classification network (Guo et al., 2017). It uses a single parameter $T > 0$ to rescale logit scores before applying the softmax function to compute the class distribution. Temperature scaling is expressed as follows:

$$p_T(y = i | x) = \frac{\exp(z_i / T)}{\sum_{j=1}^{k} \exp(z_j / T)}, \quad i = 1, \ldots, k \tag{3}$$

s.t. $z_1, ..., z_k$ are the logit values obtained by applying the network to input vector $x$. The optimal temperature $T$ for a trained model can be found by maximizing the log-likelihood $\sum_t \log p_T(y_t | x_t)$ for the held-out validation dataset. Studies show that finding the optimal $T$ by directly minimizing the ECE/adaECE measures yields better calibration results (Mukhoti et al., 2020). The adaECE measure was found to be much more robust and effective for calibration than ECE. In this study we used the adaECE for both calibration and evaluation.

## 3    Unsupervised Target Domain Calibration

We first formulate the problem of calibration under distribution shift. Let $x$ denote the input to the classifier network and $y$ be its label. We are given a labeled source domain validation-set dataset, denoted as $\mathcal{S} = \{(x_s^i, y_s^i)\}_{i=1}^{n_s}$ with $n_s$ samples, and an unlabeled target domain dataset $\mathcal{T} = \{x_t^i\}_{i=1}^{n_t}$ with $n_t$ samples. Adapting the network trained on the source domain to the target domain in an unsupervised manner without access to the labels can be achieved using various methods. Here, our goal is to calibrate the confidence of the adapted network prediction on samples from the target domain. For the sake of simplification, the adapted network will simply be referred to as the "network", the source domain validation set data as the "source data", and the unlabeled target domain data as the "target data".

Our method involves calibrating the adapted network directly on the target data. It is based on the observation that when calibrating by minimizing the adaECE score, it is unnecessary to know whether each individual prediction is correct. Instead, we only need to know the mean accuracy for each bin. Fortunately, there are techniques which, given a trained network, can estimate the network accuracy on data samples from a new domain without access to their labels (Deng & Zheng, 2021; Guillory et al., 2021; Garg et al., 2022). In this study we use the accuracy estimation method described in (Deng & Zheng, 2021). Their method suggests learning a dataset-level regression

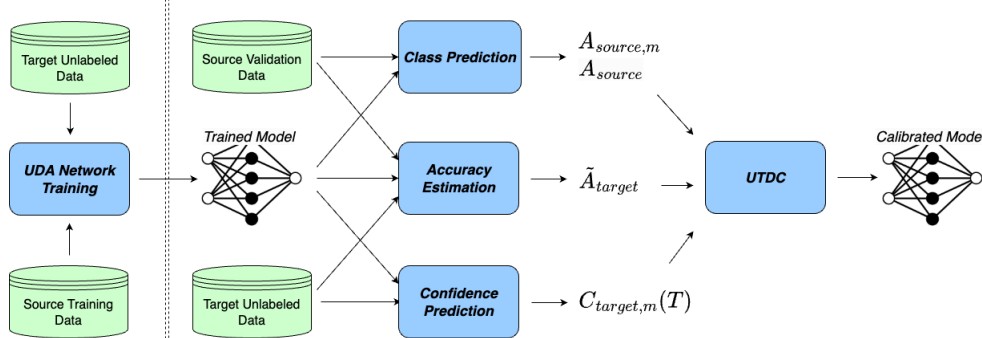

Figure 1: A scheme of the UTDC Calibration Framework.

problem. First, augment the source domain validation set, denoted by $D_s$, using various visual transformations such as resizing, cropping, horizontal and vertical flipping, Gaussian blurring, and more. We then create $n$ meta-datasets, denoted as $D_1, ..., D_n$ (in our implementation we set $n = 50$). This process preserves the labels and we can thus compute the model's accuracy on these datasets denoted by $a_1, ..., a_n$. Each dataset $D_i$ is represented as a Gaussian distribution using its mean vector $\mu_i$ and its diagonal covariance matrix $\Sigma_i$. Let $F_i$ be the Fréchet distance (Dowson & Landau, 1982) between the Gaussian representations of $D_s$ and $D_i$. $F_i$ measures the domain gap between the original dataset $D_s$ and $D_i$. Next, a linear regression model is fitted to the data set $(F_1, a_1), ..., (F_n, a_n)$ in the form of $\hat{a} = w \cdot F + b$. Finally, the linear regression model is employed to predict the accuracy of the network on the unlabeled data from the target domain.

We next suggest a simple, intuitive, and very effective method that calibrates the network directly on the target domain. We first compute the overall network accuracy on the source data $A_{\text{source}}$ and apply (Deng & Zheng, 2021) to estimate the network accuracy on the target domain. Denote the estimated target accuracy by $\tilde{A}_{\text{target}}$. Next, we divide the source data into $M$ equal-size bins according to their confidence values and compute the corresponding network accuracy $A_{\text{source},m}$ at each bin $m$. We also divide the target data into $M$ equal-size bins according to their confidence values and estimate the binwise accuracy of the target $A_{\text{target},m}$ by rescaling the binwise accuracy on the source domain in the following way:

$$\tilde{A}_{\text{target},m} = A_{\text{source},m} \cdot \frac{\tilde{A}_{\text{target}}}{A_{\text{source}}}, \qquad m = 1, ..., M. \tag{4}$$

The estimated network accuracy on the target data $\tilde{A}_{\text{target}}$ obtained by an unsupervised adaptation is usually smaller than its accuracy on the source data $A_{\text{source}}$. Thus, through this accuracy rescaling, we obtain a more realistic estimation of the bin-wise network average accuracy on the target data. Let $C_{\text{target},m}$ be the bin-wise network average confidence values computed on the target data. Substituting the estimated accuracy term, based on the source labeled data (4) into the adaECE definition (2), yields the following adaECE measure for the target domain in a UDA setup:

$$\text{UDA-adaECE} = \frac{1}{M} \sum_{m=1}^{M} \left| \tilde{A}_{\text{target},m} - C_{\text{target},m} \right|. \tag{5}$$

For each calibration method whose parameters can be found by minimizing the adaECE measure, we can form a UDA variant in which UDA-adaECE (5) is minimized instead of adaECE (2). Examples of these calibration methods include Temperature Scaling (TS), Vector Scaling, Matrix Scaling (Guo et al., 2017), Mix-n-Match (Zhang et al., 2020), Weight Scaling (Frenkel & Goldberger, 2022), and others.

We next demonstrate the UDA calibration principle in the case of TS calibration. We can determine the temperature that minimizes the UDA-adaECE measure (5) by conducting a grid search on the possible values. Given the division of the target data into bins, we can also compute the binwise average confidence after temperature calibration by $T$ on the target $C_{\text{target},m}(T)$. We can then define

---

**Algorithm 1** Unsupervised Target Domain Calibration (UTDC)

---

**input:** A labeled validation set from the source domain, an unlabeled dataset from the target domain, and a $k$-class classifier which was adapted to the target domain.

- Compute the source accuracy $A_{\text{source}}$ and estimate the target accuracy $\tilde{A}_{\text{target}}$ using (Deng & Zheng, 2021).
- Divide the source points into $M$ equal size sets based on their confidence and compute the binwise mean accuracy: $A_{\text{source},m}, \;\; m = 1, ..., M$.
- Divide the target points into $M$ equal size sets $B_1, ..., B_M$ based on their confidence.

**for** each candidate value of $T$ **do**
   - Compute the binwise mean confidence on the target:

$$C_{\text{target},m}(T) = \frac{1}{|B_m|} \sum_{x \in B_m} \max_{i=1}^{k} \frac{\exp(z_{x,i}/T)}{\sum_{j=1}^{k} \exp(z_{x,j}/T)} \qquad m = 1, ..., M.$$

    s.t. $z_{x,1},...,z_{x,k}$ are the logit values computed by the network that is fed by $x \in B_m$.
   - Compute the adaECE score as a function of $T$:

$$\text{UDA-adaECE}(T) = \frac{1}{M} \sum_{m=1}^{M} \left| A_{\text{source},m} \times \frac{\tilde{A}_{\text{target}}}{A_{\text{source}}} - C_{\text{target},m}(T) \right|$$

**end for**
**output:** The optimal temperature: $\hat{T} = \arg\min_T \text{UDA-adaECE}(T)$

---

the following temperature-dependent adaECE scores:

$$\text{UDA-adaECE}(T) = \frac{1}{M} \sum_{m=1}^{M} \left| \tilde{A}_{\text{target},m} - C_{\text{target},m}(T) \right|. \qquad (6)$$

The optimal temperature is thus obtained by applying a grid search to find $T$ that minimizes UDA-adaECE$(T)$ (6). The proposed Unsupervised Target Domain Calibration (UTDC) algorithm is summarized in Algorithm Box 1 and a scheme of it is shown in Fig. 1.

## 4 EXPERIMENTS

In this section, we evaluate the capabilities of our UTDC technique to calibrate a network on a target domain after applying a UDA procedure.

**Compared methods.** We compared our method to six baselines: (1) Uncalibrated - The adapted classifier as is, without any post-hoc calibration; (2-4) Source-TS, Source-VS and source-MS - The adapted network was calibrated by either Temperature Scaling (TS) Vector Scaling (VS) or Matrix scaling (MS) (Guo et al., 2017) using the labeled validation set of the source domain; (5) CPCS (Park et al., 2020), and (6) TransCal (Wang et al., 2020), importance weighted UDA calibrators. We also report Oracle results where the calibration was applied to the labeled data from the target domain (denoted by Target-TS) and an Oracle version of our approach (denoted by UTDC(*)) where we used the exact accuracy of the adapted model on the target data instead of estimating it. We used the adaECE measure computed on the test set of the target domain to evaluate calibration performance.

**Datasets.** We report experiments on two real-world domain adaptation benchmarks, Office-home (Venkateswara et al., 2017) and Office-31 (Saenko et al., 2010). Office-home includes four domains - Art, Real-World, Clipart and Product, represented as A, R, C, and P in the experiments. Office-31 contains three domains - Amazon, Webcam and DSLR, denoted as A, W, D.

**Implementation details.** We followed the experiment setup of Wang et al. (2020) and used their code to implement CPCS and TransCal baselines. Follwoing (Wang et al., 2020), we implemented three different UDA techniques; namely, DANN (Ganin et al., 2016), DANN+E and CDAN+E (Long et al., 2018). The performance of more recent UDA models (e.g. (Liang et al., 2021; Jin et al., 2020; Cui et al., 2020)) on the target domain of the evaluated datasets is slightly better but it is still much worse than the performance on the source domain. We provide a code implementation

Table 2: Adaptive ECE for top-1 predictions (in %) of Office-home, using 15 bins (with the lowest in bold ) on various UDA classification tasks and models with different calibration methods.

| UDA | Method | $A \to R$ | $A \to C$ | $A \to P$ | $C \to R$ | $C \to P$ | $C \to A$ | $P \to R$ | $P \to C$ | $P \to A$ | Avg |
|---|---|---|---|---|---|---|---|---|---|---|---|
| | Uncalibrated | 22.23 | 42.62 | 30.49 | 25.18 | 28.25 | 33.69 | 20.32 | 40.46 | 38.85 | 31.34 |
| | Source-TS | 8.09 | 24.43 | 14.89 | 10.00 | 14.17 | 13.85 | 11.14 | 27.42 | 26.60 | 16.73 |
| | Source-VS | 10.54 | 27.54 | 19.51 | 12.12 | 14.65 | 15.78 | 11.27 | 31.55 | 27.46 | 18.94 |
| | Source-MS | 28.62 | 47.87 | 35.74 | 31.62 | 31.54 | 40.43 | 23.59 | 43.90 | 40.56 | 35.99 |
| | CPCS | 15.84 | 49.78 | 23.42 | 14.02 | 16.60 | 18.45 | 6.31 | 49.21 | 25.62 | 24.36 |
| CDAN+E | TransCal | 6.01 | 27.30 | 9.46 | 16.67 | 16.81 | 21.69 | 19.90 | 41.23 | 39.71 | 22.09 |
| | UTDC | 4.46 | 9.74 | 7.53 | 8.36 | 5.91 | 8.08 | 10.45 | 7.46 | 9.37 | **7.93** |
| | UTDC(*) | 4.30 | 5.93 | 7.41 | 7.85 | 4.62 | 10.16 | 10.76 | 4.55 | 9.54 | 7.24 |
| | Target-TS | 3.97 | 5.05 | 7.19 | 4.07 | 4.39 | 7.07 | 2.32 | 4.39 | 8.57 | 5.22 |
| | Uncalibrated | 19.90 | 39.19 | 26.75 | 24.47 | 26.33 | 33.53 | 20.25 | 40.06 | 39.25 | 29.97 |
| | Source-TS | 6.90 | 19.80 | 7.93 | 6.54 | 7.01 | 16.01 | 15.68 | 27.87 | 30.97 | 15.41 |
| | Source-VS | 10.15 | 25.83 | 15.31 | 12.13 | 10.70 | 17.90 | 14.69 | 32.40 | 31.64 | 18.97 |
| | Source-MS | 30.78 | 52.03 | 38.39 | 35.44 | 35.45 | 44.21 | 26.40 | 45.87 | 43.33 | 39.10 |
| | CPCS | 13.90 | 50.16 | 21.32 | 3.62 | 7.25 | 34.74 | 25.86 | 22.66 | 27.97 | 23.05 |
| DANN+E | TransCal | 7.21 | 27.42 | 12.36 | 17.81 | 15.43 | 29.93 | 24.64 | 46.61 | 45.83 | 25.25 |
| | UTDC | 4.14 | 5.86 | 5.47 | 10.28 | 3.89 | 6.67 | 15.33 | 5.70 | 12.65 | **7.78** |
| | UTDC(*) | 2.68 | 4.70 | 4.37 | 8.55 | 4.00 | 4.53 | 14.60 | 3.97 | 6.16 | 5.95 |
| | Target-TS | 2.68 | 2.76 | 3.67 | 2.24 | 3.16 | 2.99 | 1.15 | 1.62 | 4.55 | 2.76 |
| | Uncalibrated | 16.82 | 31.28 | 23.11 | 17.22 | 20.46 | 27.38 | 15.88 | 33.81 | 30.13 | 24.01 |
| | Source-TS | 6.33 | 16.41 | 13.22 | 2.83 | 5.00 | 15.82 | 10.91 | 29.09 | 23.61 | 13.69 |
| | Source-VS | 10.03 | 25.58 | 15.86 | 8.10 | 8.23 | 15.18 | 11.86 | 33.08 | 27.24 | 17.24 |
| | Source-MS | 31.61 | 50.68 | 41.31 | 34.23 | 36.48 | 44.23 | 25.49 | 44.75 | 40.17 | 38.77 |
| | CPCS | 8.89 | 33.56 | 19.99 | 25.29 | 9.62 | 12.82 | 16.87 | 27.49 | 45.93 | 22.27 |
| DANN | TransCal | 7.63 | 29.15 | 22.20 | 22.64 | 22.97 | 37.66 | 26.11 | 50.85 | 47.53 | 29.64 |
| | UTDC | 5.15 | 4.87 | 11.24 | 8.63 | 5.23 | 15.08 | 18.62 | 12.62 | 11.23 | 10.30 |
| | UTDC(*) | 2.80 | 5.49 | 6.21 | 6.20 | 3.38 | 3.44 | 12.61 | 5.00 | 4.67 | 5.53 |
| | Target-TS | 2.45 | 2.38 | 4.65 | 2.08 | 1.73 | 2.16 | 1.22 | 2.35 | 2.92 | 2.44 |

Table 3: Adaptive ECE for top-1 predictions (in %) on Office-31 using 15 bins (with the lowest in bold) on various UDA classification tasks and models with different calibration methods.

| UDA Method | Method | $A \to W$ | $A \to D$ | $W \to A$ | $W \to D$ | $D \to A$ | $D \to W$ | Avg |
|---|---|---|---|---|---|---|---|---|
| | Uncalibrated | 11.5 | 10.53 | 29.63 | 1.21 | 29.08 | 1.33 | 13.88 |
| | Source-TS | 6.03 | 7.43 | 33.21 | 0.86 | 27.25 | 2.12 | 12.82 |
| | Source-VS | 3.74 | 7.10 | 33.75 | 1.52 | 32.98 | 1.42 | 13.42 |
| | Source-MS | 12.15 | 16.72 | 30.76 | 1.02 | 29.99 | 1.38 | 15.34 |
| CDAN+E | CPCS | 9.67 | 12.66 | 33.47 | 1.11 | 28.16 | 2.18 | 14.54 |
| | TransCal | 3.78 | 9.45 | 34.43 | 1.27 | 33.68 | 1.56 | 14.03 |
| | UTDC | 4.19 | 5.18 | 5.15 | 1.20 | 5.14 | 2.18 | **3.84** |
| | UTDC(*) | 3.82 | 5.18 | 5.09 | 1.13 | 25.36 | 2.18 | 7.13 |
| | Target-TS | 3.44 | 4.67 | 3.32 | 0.75 | 3.20 | 0.89 | 2.71 |
| | Uncalibrated | 13.05 | 13.55 | 28.29 | 0.87 | 27.15 | 1.68 | 14.10 |
| | Source-TS | 5.18 | 9.29 | 26.93 | 1.31 | 26.44 | 2.44 | 11.93 |
| | Source-VS | 4.63 | 8.24 | 36.64 | 0.87 | 31.35 | 1.55 | 13.88 |
| | Source-MS | 18.01 | 14.02 | 31.10 | 1.09 | 28.51 | 1.51 | 15.71 |
| DANN+E | CPCS | 15.58 | 6.81 | 33.97 | 1.99 | 32.69 | 1.14 | 15.36 |
| | TransCal | 7.98 | 5.63 | 34.53 | 1.57 | 31.12 | 1.59 | 13.74 |
| | UTDC | 5.25 | 5.33 | 8.99 | 1.40 | 12.26 | 2.41 | **5.94** |
| | UTDC(*) | 4.87 | 6.10 | 6.86 | 1.40 | 6.53 | 2.44 | 4.70 |
| | Target-TS | 3.98 | 4.77 | 2.87 | 0.85 | 2.80 | 0.82 | 2.68 |
| | Uncalibrated | 10.66 | 12.59 | 23.03 | 1.77 | 24.43 | 2.93 | 12.57 |
| | Source-TS | 3.89 | 7.17 | 29.58 | 0.98 | 30.71 | 4.43 | 12.79 |
| | Source-VS | 3.88 | 7.64 | 34.50 | 1.44 | 32.31 | 2.84 | 13.77 |
| | Source-MS | 21.06 | 24.70 | 28.81 | 1.35 | 28.45 | 1.30 | 17.61 |
| DANN | CPCS | 16.96 | 10.10 | 33.69 | 2.61 | 35.39 | 4.80 | 17.26 |
| | TransCal | 10.36 | 15.62 | 87.02 | 2.31 | 45.79 | 6.00 | 27.85 |
| | UTDC | 3.71 | 8.70 | 5.14 | 2.61 | 9.26 | 5.23 | **5.78** |
| | UTDC(*) | 5.04 | 7.52 | 5.54 | 2.61 | 12.25 | 6.54 | 6.58 |
| | Target-TS | 3.53 | 4.12 | 2.79 | 0.97 | 3.19 | 1.94 | 2.76 |

of our method for reproducibility https://anonymous.4open.science/r/unsupervised-target-domain-calibration.

**Calibration results.** Tables 2 and 3 report the calibration results on Office-home and Office-31 respectively. The results show that UTDC achieved significantly better results than the baseline methods on both tasks. The calibration obtained by previous IW-based methods was slightly better (and in some cases even worse) than a network with no calibration or a network that was calibrated on the source domain. In contrast, the adaECE score obtained by UTDC was almost as good as the adaECE obtained by an oracle that had access to the labels of the domain samples. In addition to the adaECE evaluation measure, Table 4 evaluates the average calibration results over all Office-home tasks, using three other calibration metrics as follows, ECE, Negative Log-Likelihood (NLL) and Brier Score (BS) (Brier, 1950). We can see there the same trends.

Table 4: calibration metrics results of various UDA calibration methods on the Office-home tasks.

| method | CDAN+E BS | NLL | ECE | DANN+E BS | NLL | ECE | DANN BS | NLL | ECE |
|---|---|---|---|---|---|---|---|---|---|
| Uncalibrated | 0.74 | 3.40 | 31.32 | 0.76 | 3.07 | 29.92 | 0.75 | 2.75 | 24.08 |
| Source-TS | 0.65 | 2.18 | 16.79 | 0.67 | 2.21 | 15.40 | 0.71 | 2.37 | 13.71 |
| CPCS | 0.71 | 3.48 | 24.46 | 0.72 | 3.08 | 23.12 | 0.76 | 2.87 | 22.37 |
| TransCal | 0.69 | 2.70 | 22.12 | 0.73 | 3.08 | 25.22 | 0.81 | 3.72 | 29.71 |
| UTDC | **0.62** | **1.95** | **8.01** | **0.64** | **2.01** | **7.81** | **0.69** | **2.26** | **10.35** |
| UTDC(*) | 0.62 | 1.95 | 7.21 | 0.63 | 1.99 | 5.94 | 0.68 | 2.18 | 5.53 |
| Target-TS | 0.61 | 1.92 | 5.41 | 0.63 | 1.96 | 2.72 | 0.68 | 2.14 | 2.78 |

Table 5: Computed temperature on various UDA Office-home tasks, and calibration methods using CDAN+E.

| UDA | Method | $A \to R$ | $A \to C$ | $A \to P$ | $C \to R$ | $C \to P$ | $C \to A$ | $P \to R$ | $P \to C$ | $P \to A$ | Avg |
|---|---|---|---|---|---|---|---|---|---|---|---|
| | Source-TS | 1.96 | 2.02 | 2.02 | 1.87 | 1.90 | 2.06 | 1.63 | 1.72 | 1.68 | 1.87 |
| | CPCS | 1.46 | 0.57 | 1.49 | 1.68 | 1.75 | 2.05 | 1.93 | 0.50 | 1.73 | 1.46 |
| | TransCal | 2.12 | 1.86 | 2.39 | 1.50 | 1.74 | 1.62 | 1.03 | 0.96 | 0.95 | 1.57 |
| CDAN+E | UTDC | 2.27 | 2.90 | 2.91 | 1.97 | 2.44 | 2.54 | 1.67 | 2.93 | 2.89 | 2.50 |
| | UTDC(*) | 2.29 | 3.21 | 2.68 | 2.00 | 2.62 | 2.30 | 1.65 | 3.41 | 2.90 | 2.56 |
| | Target-TS | 2.36 | 3.61 | 2.73 | 2.42 | 2.73 | 2.81 | 2.24 | 3.49 | 3.37 | 2.86 |

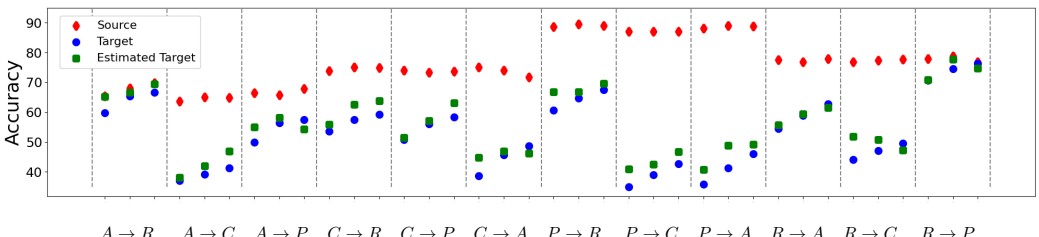

Figure 2: Average accuracy on Office-home tasks for the three UDA techniques (DANN, DANN+E, CDAN+E).

We next illustrate and analyze several key features of the proposed method.

**Accuracy gap between source and target.** To gain a better understanding of the reasons why our method performs better than IW based methods, we first show the accuracy of the adapted models on the source and target domains. Fig. 2 presents the accuracy on the source and target domains for three UDA techniques. It shows that even after adaptation to the target, the model's performance on the source samples is consistently better than its performance on the target samples, especially in cases of large domain gaps. Hence, using the network accuracy on the source to estimate the network's accuracy on the target while minimizing the ECE measure, is misleading. The over-optimistic accuracy estimation leads to a scaling temperature that is too small. Table

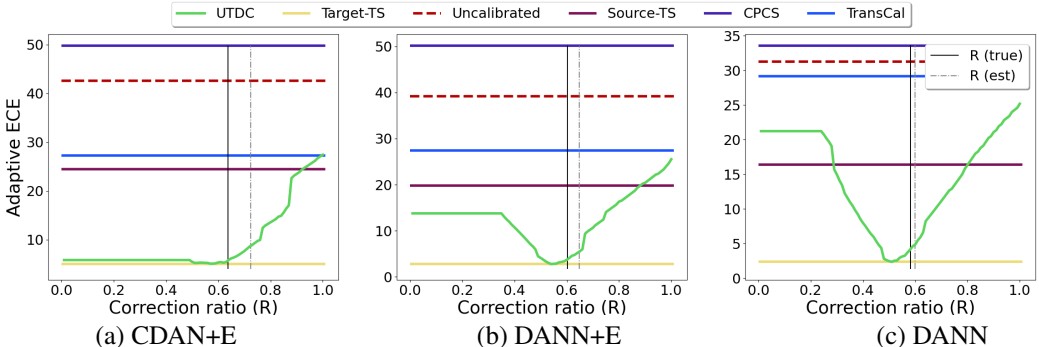

Figure 3: adaECE results as a function of the correction ratio $R$ on Office-Home, $A \rightarrow C$ task.

5 compares the optimal temperatures computed by the calibration methods. In all the baseline methods the computed calibration temperature was lower than the optimal value. This results in a poorer calibration performance as seen in Tables 2 and 3. By contrast, the temperature computed by all the UTDC variants was much closer to the optimal temperature computed by the Oracle method which had access to the target labels. Fig. 2 also presents the estimated accuracy of the adapted model on the target domain. This estimation is close to the true accuracy. Thus, when it is combined with the confidence computed on the target domain, we can obtain a calibrated mode.

**Sensitivity of UTDC to the target accuracy prediction.** UTDC is based on estimating the binwise average network accuracy on the target domain data from the labeled source domain data. This estimation is done by computing the ratio $\tilde{A}_{\text{target}}/A_{\text{source}}$ between the estimated target accuracy and the source accuracy. We next analyze the sensitivity of our calibration method to errors in estimating $A_{\text{target}}$. Let $R(\text{true}) = A_{\text{target}}/A_{\text{source}}$ and $R(\text{estimated}) = \tilde{A}_{\text{target}}/A_{\text{source}}$ be the true and estimated ratio used by UTDC(*) and UTDC respectively. In principle, every number $0 < R$ can be used to obtain an estimation of the binwise target accuracy: $\tilde{A}_{\text{target},m} = A_{\text{source},m} \cdot R$. We can thus find the temperature that minimizes the adaECE function on the target data as a function of $R$: $\hat{T}(R) = \arg\min_T \text{adaECE}_R(T)$ where

$$\text{adaECE}_R(T) = \frac{1}{M} \sum_{m=1}^{M} |A_{\text{source},m} \cdot R - C_{\text{target},m}(T)|.$$

Fig. 3 shows the adaECE measure on the target data after temperature scaling by $\hat{T}(R)$ as a function of the ratio $R$ for the task Office-home $A \rightarrow C$. It shows that with the appropriate choice of $R$ we can achieve the calibration level of the Oracle TS-target algorithm (the case in which target labels are known). This means that the accuracy difference is indeed the main reason for the calibration degradation caused by methods that try to calibrate the target domain using the source data. Specifically, as the ratio $R$ drops towards $R(\text{true})$, the adaECE improves and approaches the Oracle TS-target calibration. In addition, the adaECE reaches a minimum near $R(\text{true})$ and $R(\text{estimated})$. Finally, there is a range of correction ratios in which UTDC is better by a large margin as compared to other baselines, thus providing a tolerance for error and resilience in estimating $\tilde{A}_{\text{target}}$.

**The problem with the IW assumption.** We showed that our method achieves better results by explicitly addressing the accuracy gap between source and target domains caused by the domain shift. Previous methods based on importance weights Park et al. (2020); Wang et al. (2020), relied on re-weighting source data based on their proximity to the target data, i.e., concentrating on source samples that resemble the target and attributing less attention to others. We computed the target similarity weights associated with each sample in the source validation set and divided them into 20% percentile subsets. Fig 4 shows the average accuracy of each group and the average target accuracy. It shows that the source accuracy is similar in all bins regardless of the similarity to the target. Thus the IW assumption that source samples that are classified as targets are more relevant for calibrating the target prediction is wrong.

**Accuracy ratio across bins.** Our method computes $\tilde{A}_{target,m}$ by re-scaling $A_{source,m}$ with the same ratio for all bins, as defined in Eq. 4. This estimation is based on the assumption that the

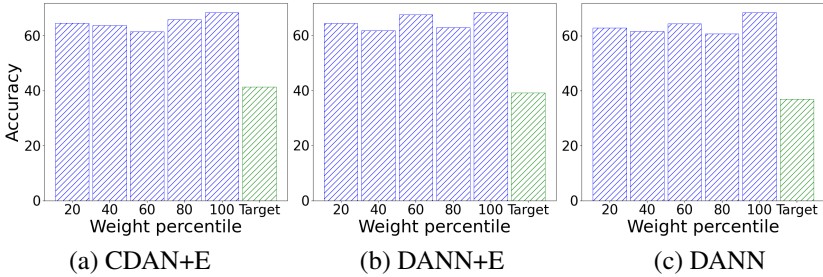

Figure 4: Accuracy of $k$-th percentile source images based on their probability to be classified as target Wang et al. (2020), compared to target accuracy (Office-home, $A \rightarrow C$).

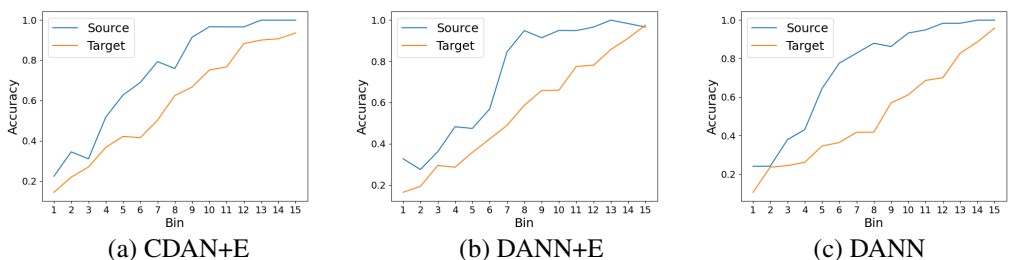

Figure 5: Accuracy per bin for source and target images. The results are shown on the Office-home $C \rightarrow P$ task.

accuracy ratio between source and target is similar across the bins. To illustrate the validity of this assumption, Fig. 5 shows the accuracy of the adapted network at each bin, for the source and target data.

## 5  CONCLUSION

To conclude, in this work, we considered the problem of network calibration in an unsupervised domain adaptation setup. We first saw that the main problem with calibration using the labeled data from the source domain is the accuracy difference between the domains. We then shawed that methods which are based on importance weighting do not address this problem, which causes them to fail. Our key idea with respect to previous methods is adapting the over-optimistic accuracy estimation performed on the labeled data from the source domain to the actual accuracy of the adapted model on the target domain, and calibrating directly over the target examples. We compared this solution to previous methods and showed that we consistently and significantly improved the calibration results on the target domain. We concentrated here on parametric calibration methods of classification tasks under domain shift. Possible future research directions are applying similar strategies to domain shift problems in regression and segmentation tasks and to domain shift problems in non-paramteric calibration methods such as conformal prediction.

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
