# OpenReview forum: "Network calibration under domain shift based on estimating the  target domain accuracy"
_ICLR.cc/2024/Conference — Submitted to ICLR 2024_

### Official Review · Reviewer_kS8x · 2023-10-27

**Soundness:** 1 poor
**Presentation:** 2 fair
**Contribution:** 2 fair
**Rating:** 3
**Confidence:** 5

**Summary:**

This submission tackles the challenge of network confidence calibration within the domain of unsupervised domain adaptation (UDA). The authors introduce a calibration framework known as Unsupervised Target Domain Calibration (UTDC). In its initial stage, UTDC applies an existing accuracy estimation technique to gauge the network's accuracy on the target domain. Subsequently, UTDC employs an existing post-hoc calibration method to directly align the network's confidence with the estimated accuracy on the target domain. The authors have conducted calibration experiments involving two distinct UDA methods across two UDA datasets. Their findings demonstrate the superior performance of UTDC when compared to existing calibration methods based on importance weighting.

**Strengths:**

1. Calibrating deep networks under domain shifts is a very important but challenging issue for the safety of deep learning applications, yet it hasn't received enough research attention.

2. The writing is easy to follow. Especially, the introduction of the network calibration problem is comprehensive and detailed.

3. Experiments show that the proposed UTDC method performs significantly better than previous calibration methods based on importance weighting.

**Weaknesses:**

1. The technical novelty is quite limited as the UTDC method is essentially a straightforward combination of existing techniques. As the authors themselves acknowledge, UTDC utilizes an established accuracy estimation method, for instance, [A], in the first step to estimate network accuracy on the target domain. In the second step, UTDC employs a well-established post-hoc calibration method, like [B], to calibrate the network using target data.

2. The empirical evaluation is rather limited and, as a result, cannot sufficiently demonstrate the soundness, effectiveness, and generalizability of the proposed method. Reasons are: (1) The evaluation solely includes DANN and CDAN as UDA methods, both of which are founded on adversarial domain adaptation and cannot sufficiently represent the broader spectrum of UDA algorithms. It is necessary to incorporate various UDA methods, including those mentioned in the "implementation details" section by the authors, such as [C, D, E, F]. (2) The experiments are exclusively conducted on the Office-Home and Office-31 datasets, neglecting large-scale UDA datasets like VisDA [G] and DomainNet [H]. (3) The calibration experiments are limited to only two importance-weighting methods, without considering significant baselines like MC-Dropout [I] or Ensemble [J]. (4) The study lacks experiments exploring the replacement of the target-accuracy estimation method or the post-hoc calibration method with other choices, which could provide valuable insights.

3. The proposed UTDC method lacks practicality for real-world UDA applications due to its complexity and limitations. A key aspect of UTDC is the use of the target accuracy estimation method [A] to estimate accuracy on the target domain. However, this method has several practical issues: (1) It involves the complex construction and modeling of numerous datasets, which can be computationally demanding, potentially exceeding the computational requirements of UDA model training. (2) It requires multiple method-specific hyperparameters, which can be challenging to determine in UDA settings. (3) It heavily depends on source data and is not applicable to more practical UDA scenarios, such as partial-set UDA [K] and source-free UDA [L].

References:

[A] Are labels always necessary for classifier accuracy evaluation? CVPR 2021

[B] On calibration of modern neural networks. ICML 2017

[C] Domain adaptation with auxiliary target domain-oriented classifier. CVPR 2021

[D]  Minimum class confusion for versatile domain adaptation. ECCV 2020

[E] Towards discriminability and diversity: Batch nuclear-norm maximization under label insufficient situations. CVPR 2020

[F] Larger norm more transferable: An adaptive feature norm approach for unsupervised domain adaptation. ICCV 2019

[G] Visda: The visual domain adaptation challenge. 2017

[H] Moment matching for multi-source domain adaptation. ICCV 2019

[I] Dropout as a bayesian approximation: Representing model uncertainty in deep learning. ICML 2016

[J] Simple and scalable predictive uncertainty estimation using deep ensembles. NeurIPS 2017

[K] Partial transfer learning with selective adversarial networks. CVPR 2018

[L] Unsupervised domain adaptation without source data. CVPR 2020

**Questions:**

1. The primary questions have been outlined in the weaknesses section.

2. Minor suggestions:
(1). I recommend adding a detailed introduction to the target domain accuracy estimation method because it is a crucial component of the method.
(2). I recommend refining the writing to ensure it is well-organized and comprehensive. For example, in the "Conclusion" section, please correct "shawed" to "showed," and consider starting the "References" section on a new page.

---

### Official Review · Reviewer_YiVF · 2023-10-31

**Soundness:** 2 fair
**Presentation:** 3 good
**Contribution:** 2 fair
**Rating:** 5
**Confidence:** 4

**Summary:**

The paper proposes a method to calibrate the predicted probabilities of an adapted model to the target domain. Since the data from the target domain is unlabelled, it is difficult and not clear on how to obtain the target (domain) accuracy. Reliable target accuracy is required to accurately quantify the calibration performance via calibration metrics and optimize such metrics to find optimal scaling parameters for post-hoc calibration methods. The paper points out the limitations of importance weighting (IW) methods and directly estimates the target accuracy by leveraging an existing technique and use it to scale the bin-wise source accuracy for achieving the bin-wise target accuracy. This bin-wise target accuracy is then plugged into the adaptive ECE (adaECE) metric and this adaECE metric is optimized in order to search for optimal temperature scaling method. Experiments are computed on two different datasets: Office-Home and Office31 to validate the calibration performance of the proposed method. Results claim to surpass the existing the methods by significant margins.

**Strengths:**

- Calibrating the predicted confidences of the adapted model to the target domain is a relevant problem because it will enhance the trust in the predictions of such models and will increase their adoption in real-world scenarios.

- The proposed idea of estimating the target accuracy and using it scale the bin-wise source accuracy to yield bin-wise target accuracy is simple and effective.

- Results on two different domain adaptation datasets show that the method outperforms competing methods by visible margins.

- The paper provides studies on different components of the method, including the role of ratio (R) used to scale the bin-wise source accuracy.

**Weaknesses:**

- The overall contribution of the paper seems a bit limited primarily because the proposed method uses the previously published method of [Deng & Zheng 2021] to obtain the target accuracy, which is a core part of proposed method.

- The experimental evaluation is not exhaustive as only two datasets: office-home and office-31 are used to show calibration results.

- The results are mostly reported with adaECE and it would be interesting to see the results with static calibration error (SCE), which takes into account the whole confidence vector calibration performance.

- The results are shown for rather similar shift scenarios; it would be interesting to see results on more varied scenarios such as presented in ViSDA2017 dataset [A].

- Why these UDA methods (DANN DANN+E, and CDAN+E) was chosen to implement the proposed calibration technique?

- The scope of the evaluation also seems limited since results were only shown for same architecture (i.e. CNNs), image datasets (two DA datasets), and only classification task.

- The compared methods are not very recent (i.e. from 2020).


[A] Peng, X., Usman, B., Kaushik, N., Wang, D., Hoffman, J. and Saenko, K., 2018. Visda: A synthetic-to-real benchmark for visual domain adaptation. In Proceedings of the IEEE Conference on Computer Vision and Pattern Recognition Workshops (pp. 2021-2026)

**Questions:**

- Can the computed target accuracy be also effective with other popular calibration metrics e.g., Expected Calibration Error (ECE)?

- It would be interesting to see the confidence histogram of incorrect predictions.

**Details Of Ethics Concerns:**

No ethics concerns.

---

### Official Review · Reviewer_LtNi · 2023-10-31

**Soundness:** 3 good
**Presentation:** 3 good
**Contribution:** 1 poor
**Rating:** 3
**Confidence:** 4

**Summary:**

This paper proposes a post-hoc uncertainty calibration method (based on temperature scaling) under domain shift setting. The key idea of the proposed method is based on the observation of previous studies, i.e., the accuracy of model on target domain can be estimated based on solving a simple regression problem. Then, this paper directly uses the estimated accuracy of each bin as the expected confidence and searches the optimal temperature to scale the confidence.

**Strengths:**

1. The studied problem of calibrating model under domain shift setting is important.

2. The motivation that the estimated target accuracy can be used to calibrate confidence is clear and sound.

**Weaknesses:**

1. The proposed approach seems a very simple combination of the idea of target accuracy estimation and model calibration. There is no deeper investigation or theoretical analysis on how to use the correlation between source domain accuracy and target domain accuracy to better calibrate the model.

2. The experimental setting is somehow limited, especially on the number of datasets. There are some other common datasets in domain adaptation and model calibration such as cifar/cifar-c can be used in this study.

3. Some references, such as [Accuracy on the Line: On the Strong Correlation
Between Out-of-Distribution and In-Distribution Generalization, ICML 2021] and [Confidence and Dispersity Speak: Characterising Prediction Matrix for Unsupervised Accuracy Estimation, ICML 2023] on studying target domain accuracy estimation or the correlation between the model performance in different domains are missed.

**Questions:**

Please refer to weaknesses section.

---

### Official Review · Reviewer_fEXd · 2023-10-31

**Soundness:** 2 fair
**Presentation:** 2 fair
**Contribution:** 1 poor
**Rating:** 3
**Confidence:** 5

**Summary:**

This work addresses calibration challenges encountered during distribution shifts, particularly in unsupervised domain adaptation (UDA). The primary focus is to calibrate model predictions in the target domain without the need for labels. The main idea is to estimate target accuracy and minimize the disparity between the estimated accuracy and the computed confidence. Based on this, the optimal temperature is identified to minimize metrics like ECE or adaECE. Tests on datasets like Office-Home and Office-31 show the proposed method achieves some improvements over several baselines.

**Strengths:**

- Relevance of Task: The emphasis on calibration under distribution shift is significant, especially given that most existing works focus primarily on in-distribution settings.

- Informative Comparison: Table 1 effectively differentiates the proposed method from its counterparts, making it easier to discern its uniqueness and potential advantages.

**Weaknesses:**

- [*Motivation Ambiguity*] A calibrated model's prediction confidence should match its dataset accuracy. Why calibrate if we can just estimate the accuracy? Isn't the estimated accuracy alone sufficient for understanding model uncertainty?

- [*Method Questionability*] Eq.4 assumes similar over/under confidence across bins to estimate each bin's accuracy. This assumption needs further justification. Why is it valid?

- [*Accuracy Estimation Concerns*] Using (Deng & Zheng, 2021)) for accuracy estimation could be problematic for large datasets or significant distribution shifts. The reliability of this method, especially under these conditions, hasn't been thoroughly discussed. Exploring other accuracy estimation techniques might be beneficial.

- [*Limited Datasets*] The chosen datasets for experiments seem small. Evaluations on standard benchmarks like ImageNet-C, CIFAR-C, and ImageNet-S are missing, making it challenging to determine the method's true advantage.

- [*Comparison with Recent Techniques*] The manuscript could benefit from comparisons to recent calibration methods:  [a] Adaptive Calibrator Ensemble: Navigating Test Set Difficulty in Out-of-Distribution Scenarios. ICCV 2023
[b] Robust calibration with multi-domain temperature scaling. In NeurIPS 2022
[c] Beyond In-domain scenarios: robust density-aware calibration. In ICML 2023
[d] Confidence calibration for domain generalization under covariate shift. In ICCV 2021

**Questions:**

- Please comment on the reliability of the estimated accuracy. Also, the scalability of the proposed method on large-scale and real-world datasets is not clear.

- Please compare with related works and discuss the differences. Without it, it is hard to see the contribution of this work

---

### Meta-Review · Area_Chair_vBD2 · 2023-12-07

**Metareview:**

This paper proposes a post-hoc uncertainty calibration method (based on temperature scaling) under domain shift setting. The key idea of the proposed method is based on the observation of previous studies, i.e., the accuracy of model on target domain can be estimated based on solving a simple regression problem.

All three reviewers have negative views on the technical contribution of this paper. Based on the overall reviews, I am inclined to reject this paper.

**Justification For Why Not Higher Score:**

All three reviewers have negative views on the technical contribution of this paper. No response during the rebuttal phase.

**Justification For Why Not Lower Score:**

N/A.

---

### Decision · Program_Chairs · 2024-01-16

Reject